

# Trends and biases in African large carnivore population assessments: identifying priorities and opportunities from a systematic review of two decades of research

Paolo Strampelli[1,*], Liz AD Campbell[1,*], Philipp Henschel[2], Samantha K. Nicholson[3,4], David W. Macdonald[1] and Amy J. Dickman[1]

[1] Wildlife Conservation Research Unit (WildCRU), Department of Zoology, University of Oxford, Oxford, United Kingdom
[2] Panthera, New York, United States of America
[3] Endangered Wildlife Trust, Johannesburg, South Africa
[4] The University of KwaZulu-Natal, Durban, South Africa
* These authors contributed equally to this work.

Corresponding author
Paolo Strampelli,
paolo.strampelli@zoo.ox.ac.uk

## ABSTRACT

African large carnivores have undergone significant range and population declines over recent decades. Although conservation planning and the management of threatened species requires accurate assessments of population status and monitoring of trends, there is evidence that biodiversity monitoring may not be evenly distributed or occurring where most needed. Here, we provide the first systematic review of African large carnivore population assessments published over the last two decades (2000–2020), to investigate trends in research effort and identify knowledge gaps. We used generalised linear models (GLMs) and generalised linear mixed models (GLMMs) to identify taxonomic and geographical biases, and investigated biases associated with land use type and author nationality. Research effort was significantly biased towards lion (*Panthera leo*) and against striped hyaena (*Hyaena hyaena*), despite the latter being the species with the widest continental range. African wild dog (*Lycaon pictus*) also exhibited a negative bias in research attention, although this was partly explained by its relatively restricted distribution. The number of country assessments for a species was significantly positively associated with its geographic range in that country. Population assessments were biased towards southern and eastern Africa, particularly South Africa and Kenya. Northern, western, and central Africa were generally under-represented. Most studies were carried out in photographic tourism protected areas under government management, while non-protected and trophy hunting areas received less attention. Outside South Africa, almost half of studies (41%) did not include authors from the study country, suggesting that significant opportunities exist for capacity building in range states. Overall, large parts of Africa remain under-represented in the literature, and opportunities exist for further research on most species and in most countries. We develop recommendations for actions aimed at overcoming the identified biases and provide researchers, practitioners, and policymakers with priorities to help inform future research and monitoring agendas.

## INTRODUCTION

Africa is host to a unique diversity of large carnivore species, including the lion (*Panthera leo*), leopard (*Panthera pardus*), cheetah (*Acinonyx jubatus*), African wild dog (*Lycaon pictus*), spotted hyaena (*Crocuta crocuta)*, striped hyaena (*Hyaena hyaena*), and brown hyaena (*Parahyaena brunnea*). Nevertheless, large carnivore populations across Africa have undergone rapid declines in recent decades, primarily as a result of habitat loss and fragmentation, loss of prey, and conflict with humans (*Brodie, Williams & Garner, 2021*; *Wolf & Ripple, 2017*). As large carnivores can help generate tourism revenue (*Hemson et al., 2009*), often have strong cultural importance (*Dheer et al., 2021*), and play a key role in the provision of ecosystem services (*Davidson et al., 2019*), there are strong practical and intrinsic motivations for the successful conservation of remaining African large carnivore populations.

Knowledge of the status and trends of populations is essential for effective conservation management of large carnivores (*Elliot & Gopalaswamy, 2016*). Information on population status allows practitioners to identify threats, evaluate the effectiveness of interventions, implement adaptive monitoring programmes, and inform policy (*Suryawanshi et al., 2019*; *Witmer, 2005*). Indeed, data on large carnivore population status have been used in recent years to inform a wide range of management and policy decisions, including national and regional action plans and conservation strategies (*IUCN, 2007*; *TAWIRI, 2016*), range-wide meta-analyses (*Weise et al., 2017*), international policies (*USFWS, 2015*), extinction risk assessments (*IUCN, 2020*), and trophy hunting quota setting (*Mweetwa et al., 2018*; *Packer et al., 2011*). Such data have also been employed to identify population strongholds (*Riggio et al., 2013*) and range-wide conservation priorities (*Bauer et al., 2015a*), and to inform controversial management practices (*Miller & Funston, 2014*; *Packer et al., 2013*).

Conversely, lack of data on population status often leads to knowledge gaps being filled by expert opinion (*Weise et al., 2017*), which can delay or prevent conservation actions (*Artelle et al., 2013*; *Popescu et al., 2016*), or lead to inappropriate or harmful management decisions (*Darimont et al., 2018*; *Moqanaki, Jiménez & López-Bao, 2018*). Successful large carnivore conservation therefore requires reliable population assessments (*Braczkowski et al., 2020*), ideally from a wide range of geographical and management contexts. However, as a result of often existing at naturally low densities, exhibiting nocturnal and secretive traits, and ranging across large areas, estimating population parameters for large carnivores can be logistically challenging and financially costly (*Karanth & Nichols, 2017*). Indeed, African carnivore population measures have been argued to be severely lacking (*Riggio et al., 2013*), and there is evidence that biodiversity monitoring and research as a whole is under-represented in Africa (*Di Marco et al., 2017*; *Martin, Blossey & Ellis, 2012*; *Stocks et al., 2008*; *Velasco et al., 2015*).

Conservation research also suffers from both taxonomical and geographical sampling biases, with research not always targeting the areas or species with the largest knowledge

gaps (*Di Marco et al., 2017*). Indeed, research can be geographically biased not only at the country level, but also towards specific regions, ecosystems, and land use categories (*Velasco et al., 2015*). Resulting knowledge gaps can compromise the implementation of science-based conservation interventions (*Trimble & van Aarde, 2012*), and biases might translate into policies, impacting the achievement of biodiversity conservation targets (*Velasco et al., 2015*). It is therefore important to understand and monitor these biases in research, in order to re-align research priorities where needed (*Di Marco et al., 2017*; *Donaldson et al., 2016*). Given ongoing funding shortfalls in conservation (*Lindsey et al., 2018*), it is especially important that limited resources are directed toward where they are most needed (*Trimble & van Aarde, 2012*).

In the context of large carnivore population status assessments and monitoring, population density is a commonly-employed and meaningful metric (*Boitani & Powell, 2012*). Although other metrics of population status exist (*e.g.*, occupancy—*Strampelli et al., 2022a*), population density is increasingly recommended as the metric of choice to assess and monitor large carnivore populations (*Elliot & Gopalaswamy, 2016*). In addition to estimating current status, estimating densities also allows comparison over time and between sites, enabling researchers and managers to understand how population status varies with biotic factors (*Searle et al., 2021*), anthropogenic disturbances (*Balme, Slotow & Hunter, 2010*; *Henschel et al., 2011*), or land management strategies (*Swanepoel, Somers & Dalerum, 2015*).

We carry out the first systematic review of all published, peer-review studies estimating population density or abundance of large African carnivore populations over two decades (2000–2020). We determine research patterns and the geographical and taxonomic representativeness of these studies, and identify biases and data gaps in population assessments and monitoring. We also employ findings to discuss representativeness of different land use types in research efforts, and the extent of involvement of authors from host countries. Finally, we use this information to identify research priorities and opportunities for future large carnivore population assessments.

## MATERIALS AND METHODS

### Literature review

We conducted a systematic review of published, peer-reviewed literature using Google Scholar (last search: January 2021), compiling all studies where population density of one or more African large carnivore species (lion, leopard, cheetah, African wild dog, spotted hyaena, striped hyaena, brown hyaena) was either explicitly estimated, or derived from an empirically measured parameter (*e.g.*, home ranges). The review protocol was applied following the Preferred Reporting Items for Systematic Reviews and Meta-Analyses (PRISMA) guidelines (*Page et al., 2021*; see Fig. 1 for PRISMA flow diagram and Appendix S1 for PRISMA checklist). Literature was searched by P.S. and S.N. by entering, in quotation marks, full scientific and vernacular species names, as well as "density", "abundance", "population", "assessment", or "survey". Both "hyaena" and "hyena" were used in the search.

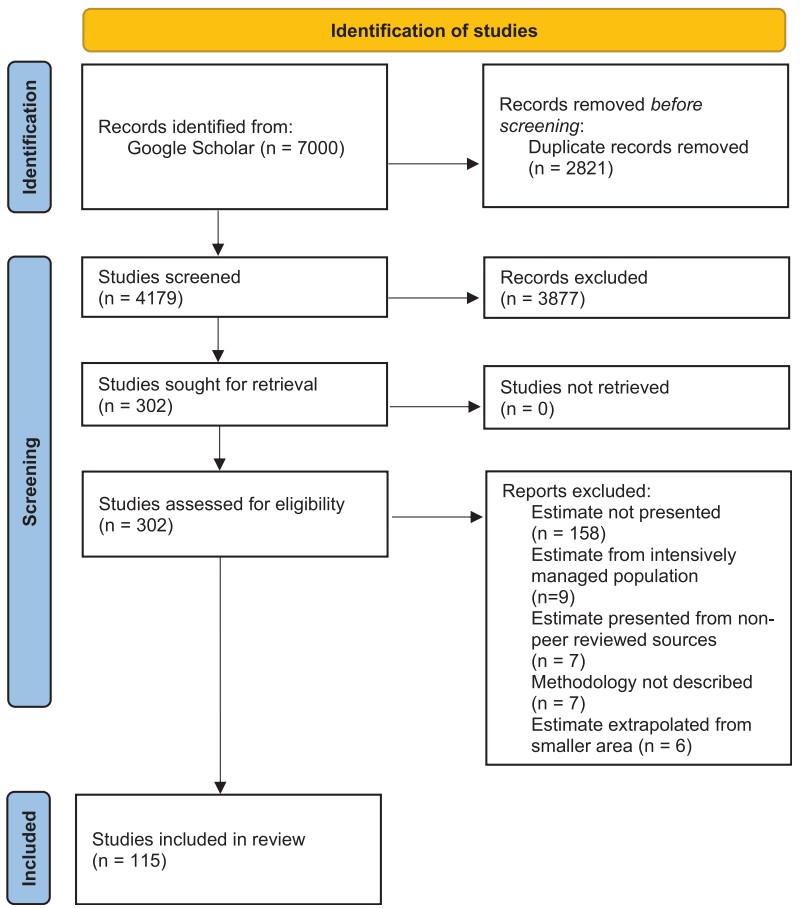

**Figure 1 PRISMA flow diagram.** PRISMA flow diagram for synthethic review of African large carnivore assessments (2000–2020).

Our intention was to review trends and biases in peer-reviewed literature over the past two decades, and in doing so help identify priorities for assessments for the coming two decades. As a result, only studies published between 1st January 2000 and 31st December 2020 were considered. Due to the number of articles returned using Google Scholar searches, only the first 100 results for each search were considered. P.S. and S.N. screened the resulting records, and all papers that did not include density or abundance estimation as an output were excluded (Fig. 1). There were no disagreements on classifications.

Only estimates published in peer-reviewed literature were considered, to ensure data and information quality (*Suryawanshi et al., 2019*). In addition, due to many unpublished reports not being publicly available, studies published in the scientific literature are those most readily available to most policy makers, conservation managers, and researchers, and therefore most likely to influence international conservation policy and the disbursement of conservation funding (*Giam & Wilcove, 2012*). Finally, the fact than many such reports are not publicly available would have also led to any search for population assessment in grey literature being incomplete, and to considerable study biases.

Although it was not necessary for population density to be the primary parameter of interest in the study, studies were only included if the methodology employed to obtain

density estimates was described, thus providing the opportunity for this aspect of the study to be subjected to peer review. Studies estimating population abundance were included only if the size of the sampled area was explicitly defined and measured. Estimates reported in a peer-reviewed study from non-peer reviewed sources (*e.g.*, *Weise et al., 2017*), where the methodology employed was not described (*e.g.*, *Balme et al., 2017*), or where estimates for a wider area were obtained indirectly by extrapolating from a smaller sampled area (*e.g.*, *Trinkel, 2009*) were not included. We also did not include density estimates from intensively managed populations (*e.g.*, *Buk et al., 2018*), as the exact number of individuals being known prior to the study precluded the type of 'exploratory' surveys of interest in this review. This was only relevant to some populations of lion, cheetah, and African wild dog in South Africa.

For each study that fitted the above criteria, we extracted information on: year of publication, year(s) of data collection, authors' nationality (national of study country or foreigner, based on a web search), study area, ecosystem, country, region, land use type (as described in the publication; if multiple uses occurred—*e.g.*, photographic tourism and trophy hunting—both were listed), density estimate, and estimation method. Risk of bias was minimised by searching for multiple possible terms in Google Scholar and by the clear-cut definitions of inclusion employed.

Finally, we are aware of recent debates regarding the reliability of some large carnivore population assessment methods (*e.g.*, *Braczkowski et al., 2020*; *Dröge et al., 2020*), and of common issues in large carnivore density estimation studies, such as that of under-sampling the study area (*Suryawanshi et al., 2019*). However, given that our goal was to assess research effort, we did not make distinctions based on methods employed, or on other features of the studies themselves.

## Analyses

Findings of the literature review were employed to identify taxonomic trends in research effort, including the total number of studies and the study density (studies per km$^2$ of geographical range) per species. Geographic range maps and data for each species were obtained from the IUCN Red List (https://www.iucnredlist.org/). Although we acknowledge some of these may be outdated or imperfect, we reasoned this was the best way to remain consistent across species. We also determined spatial patterns in research, through the total number of studies and the study density per country and per region, both by species and overall. Finally, we investigated patterns in land tenure of the areas where research was carried out, and the nationality (local national *vs* foreigner) of the authors of the research.

We employed Generalised Linear Mixed Models (GLMMs) and Generalised Linear Models (GLMs) to investigate taxonomic and spatial biases in large carnivore population assessments. Models that combined species were fitted by Poisson GLMMs, with the number of studies coded as the explanatory variable. Country and Species were included as random effects (random intercepts) to control for multiple observations (*i.e.*, data points for multiple species in each country, and for multiple range countries for each species), and

to investigate research biases (see below). All models were validated by posterior predictive checks, dispersion parameters, and model residuals (Appendix S5).

To first determine overall biases in research effort by species and countries, a model was built with only Species and Country as random effects (Model 1). In this model, the random intercepts represent relative differences in the number of surveys for that species or country; positive values indicate a greater than average number of studies for that species or country, while negative values indicate a lower than average number of studies, with the magnitude indicating the degree of difference.

Species with larger geographical ranges could be expected to receive greater research attention by chance. Therefore, to control for this, we then fit a model controlling for differences in species' geographical range (Range) between countries, with Range modelled as a fixed effect and Country and Species and random effects (Model 2). This allowed us to (a) test whether greater research effort was directed towards species and countries with larger ranges; and (b) assess biases in research, by country and by species, after controlling for differences in geographical range. For this model, the random intercepts indicate whether the number of studies for a species or in a country was more or less than expected based on the range of that species or in that country.

All analyses followed a Bayesian approach to parameter estimation using JAGS (Plummer, 2003) in R (version 3.6.1; R Core Team, 2019) with the package *runjags* (Denwood, 2020). The fixed explanatory variable ('range') was standardised to have a mean of 0 and standard deviation of 1, to improve MCMC (Markov Chain Monte Carlo) convergence (Zuur, Hilbe & Leno, 2013). All models used three MCMC chains, diffuse priors, and ran for enough iterations to produce an effective sample size >10,000 (Kruschke & Liddell, 2018). MCMC chain convergence was confirmed with trace-plots and the Gelman-Rubrin statistic. Variables (both fixed and random effects) were considered to have a significant effect if they exhibited a credible non-zero effect (*i.e.*, zero not contained within the 95% highest density interval (HDI) of the posterior distribution).

In addition to the all-species GLMMs, GLMs were also built for each species to investigate species-specific research biases. GLMs were used over GLMMs as—unlike the all-species models—datasets for individual species did not have multiple observations per country, thus allowing their applicability. Country biases were assessed using the model residuals: a positive residual signified a country had more studies than expected, while a negative residual signified fewer. Residuals considered credibly different from zero were those for which zero was not contained in the 95% highest density interval (HDI).

GLMs were first fitted with a Poisson distribution (as we employed discrete count data) and validated by posterior predictive checks, dispersion parameters, and model residuals. Poisson distribution assumes that the mean and variance of the data are equal; if a Poisson model was not suitable for the data, a more complex negative binomial model was used, validated by the same methods employed for the GLMMs (Appendix S5). Through this method, leopard, spotted hyaena, and cheetah data were fitted with Poisson GLMs with one outlier removed to reduce overdispersion; lion data were fitted with a negative binomial GLM. African wild dog, striped hyaena, and brown hyaena could not be

modelled individually due to too few data points (range countries) and/or insufficient variance (*i.e.*, most range countries had zero studies).

## RESULTS

### Literature review and research trends

Our search of the published literature revealed 115 peer-reviewed articles (studies) which estimated population density of one or more African large carnivore species. These provided a total of 312 estimates of large carnivore population density, inclusive of all species (Figs. 2 and 3).

Studies were predominantly carried out on protected land managed by the government (63%), followed by private reserves (19%), land under community-based management (16%), and unprotected areas (10%). Land tenure was unclear for 2% of studies. A total of 83% of studies took place in areas where non-consumptive (photographic) tourism occurred, 33% in an area with livestock ranching, game ranching, and/or farming, 15% in an area with trophy hunting, and 3% in a logging or mining concession. 71% of studies included a national of the study country as an author. For studies outside of South Africa, however, only 59% of studies included a national of the study country as an author. See Appendix S2 for a complete list of all studies and associated information, and Appendix S3 for additional details on data interpretation.

### Taxonomic trends and biases

Lion was the species with the greatest number of population density assessments in peer-reviewed literature (55 studies which fitted the described criteria, leading to 90 estimates of population density), followed by spotted hyaena (34 studies, 81 estimates), leopard (33 studies, 71 estimates), cheetah (19 studies, 21 estimates), brown hyaena (11 studies, 39 estimates), African wild dog (six studies, seven estimates), and finally striped hyaena (three studies, three estimates) (Fig. 3). Results from the GLMM Model 1 (Fig. 4; Table S9) confirmed that more studies than average were conducted on lion (which exhibited a significant positive bias) and on spotted hyaena, leopard, brown hyaena and cheetah, and fewer than average on African wild dog and striped hyaena (with the latter exhibiting a significant negative bias). Lion was also the species with the greatest study density (studies per 100,000 $km^2$ of range), with this being an order of magnitude greater than for all other species (Fig. 3).

After controlling for differences in geographical range (GLMM Model 2), there were still significantly fewer studies than expected on striped hyaena; in fact, the species' very large geographical range (Table S1) resulted in the negative bias for the species increasing (Table 1; Fig. 4). Lion still experienced a significant positive bias, while results suggest that the negative effect for African wild dog and the positive effect for spotted hyaena were partly explained by their restricted and large geographical ranges, respectively (Table S1; Fig. 4).
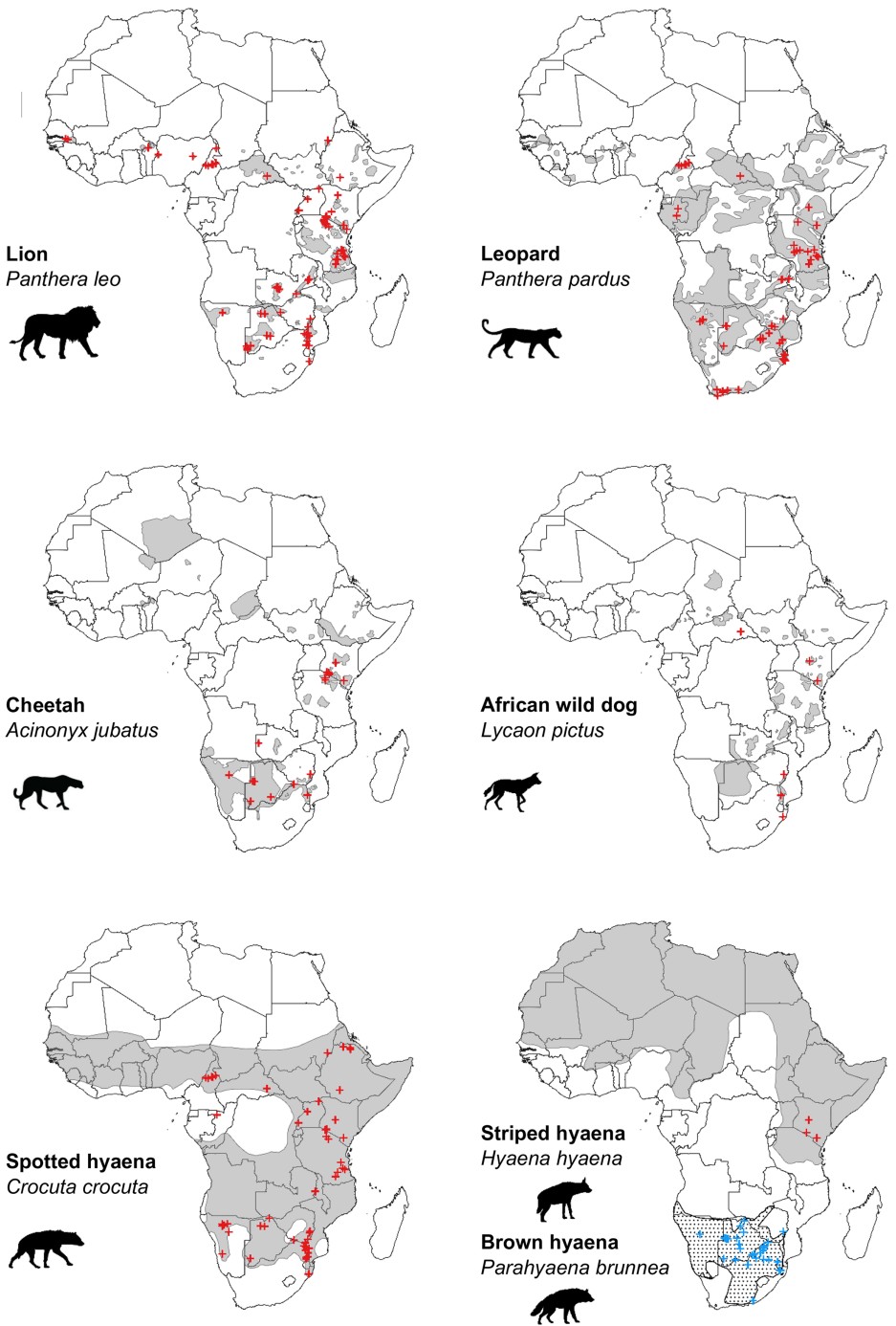

**Figure 2 Location of peer-reviewed African large carnivore studies (2000–2020).** Location of peer-reviewed African large carnivore population assessment studies (2000–2020). Red crosses show estimates of population density, except in the case of brown hyaena, where blue crosses are used. When a study estimated multiple population densities, these are shown as separate points. Grey areas represent current geographical ranges (*IUCN, 2020*), except in the case of brown hyaena, where a dotted pattern is used. Diûerent symbols were used for brown hyaena in order to facilitate presentation of the data.

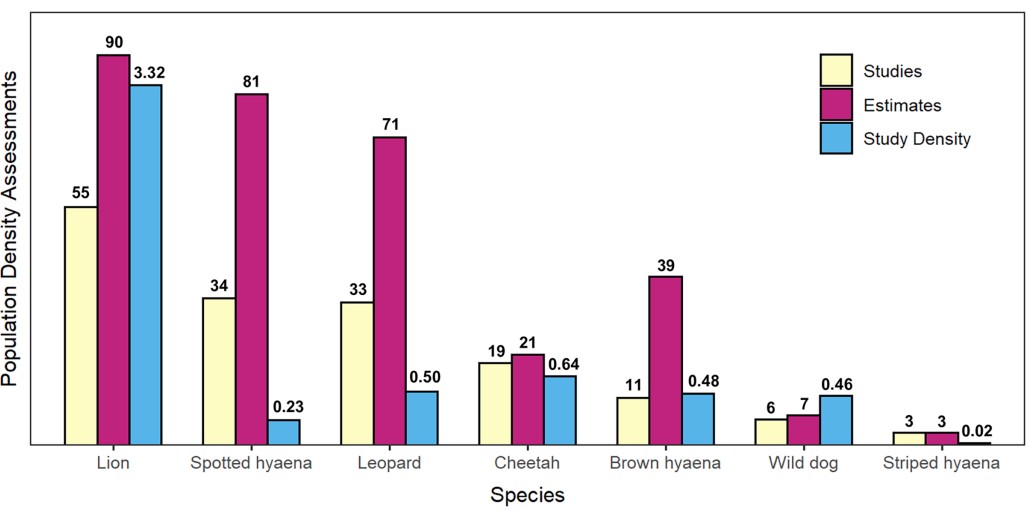

**Figure 3 Summary of peer-reviewed African large carnivore density assessments, by species.** Number of peer-reviewed population density assessments (studies), number of individual population density estimates, and density of peer-reviewed studies (studies per 100,000 km$^2$ of geographical range) for African large carnivores (2000–2020). Species are presented in decreasing order by number of studies.

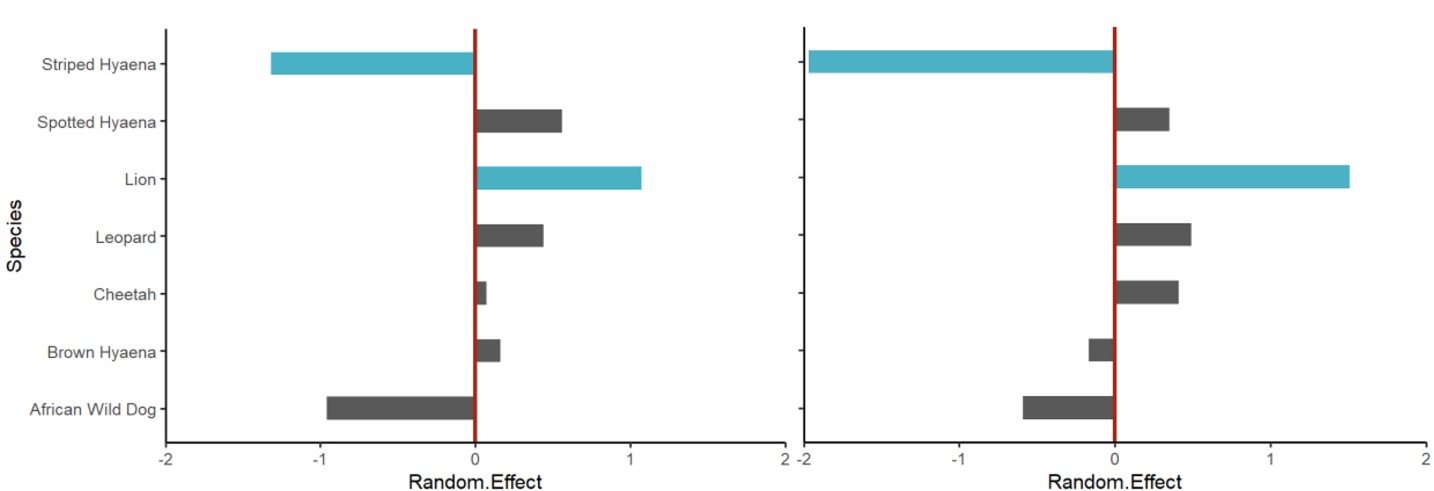

**Figure 4 Random intercept values for species from the Poisson GLMMs.** Random intercept values for species from the Poisson GLMM without explanatory variables (Model 1, Table S9) and for that accounting for geographical range (Model 2, Table 1). In both cases, values for lion and striped hyaena were credibly different from zero (*i.e.*, zero not contained within the 95% HDI of the posterior distribution; shown in blue), with this being strongest when accounting for geographical range of species. The moderate negative effect for African wild dog and positive effect for spotted hyaena, on the other hand, were partly explained by their restricted and very large geographical ranges, respectively.

## Geographical trends and biases

### Regional

Across all species, southern Africa was the region with the greatest number and density of large carnivore population assessments, followed by eastern Africa (Table 2).

**Table 1 Results of the Poisson GLMM investigating biases in large carnivore population assessments in Africa, accounting for differences in geographical range between countries (Model 2).** Species and countries are modelled as random effects. For species, a positive value indicates more assessment than expected given their geographical range, while a negative fewer. For countries, a positive value indicates more assessments than expected based on large carnivore geographical range within that country, while a negative fewer. Values credibly different from zero (*i.e.*, 95% highest density interval (HDI) of the posterior distribution does not contain zero) are highlighted in bold for species and range.

| Name | Mean | HDI low | HDI high | SD |
|---|---|---|---|---|
| Intercept | −2.07 | −3.60 | −0.65 | 0.75 |
| **Range*** | **0.60** | **0.26** | **0.93** | **0.17** |
| Country random effect variance | 1.70 | 1.07 | 2.46 | 0.37 |
| Species random effect variance | 1.46 | 0.52 | 2.78 | 0.69 |
| Brown hyaena | −0.17 | −1.53 | 1.20 | 0.69 |
| Cheetah | 0.41 | −0.91 | 1.75 | 0.67 |
| Leopard | 0.49 | −0.76 | 1.80 | 0.65 |
| **Lion*** | **1.51** | **0.29** | **2.88** | **0.66** |
| Spotted hyaena | 0.35 | −0.93 | 1.65 | 0.65 |
| **Striped hyaena[†]** | **−1.97** | **−3.82** | **−0.39** | **0.89** |
| African wild dog | −0.59 | −2.00 | 0.82 | 0.71 |
| South Africa* | 3.13 | 2.24 | 4.08 | 0.48 |
| Kenya* | 2.91 | 2.02 | 3.87 | 0.48 |
| Cameroon* | 2.47 | 1.45 | 3.56 | 0.54 |
| Botswana* | 2.38 | 1.43 | 3.38 | 0.50 |
| Tanzania* | 2.19 | 1.21 | 3.24 | 0.52 |
| Zimbabwe* | 2.10 | 1.05 | 3.20 | 0.55 |
| Namibia* | 1.68 | 0.62 | 2.78 | 0.55 |
| Zambia* | 1.48 | 0.36 | 2.64 | 0.58 |
| Mozambique | 1.04 | −0.18 | 2.28 | 0.63 |
| Uganda | 0.96 | −0.48 | 2.37 | 0.73 |
| CAR | 0.92 | −0.40 | 2.20 | 0.66 |
| Ethiopia | 0.86 | −0.38 | 2.12 | 0.63 |
| Malawi | 0.73 | −0.95 | 2.31 | 0.83 |
| Senegal | 0.61 | −1.04 | 2.18 | 0.82 |
| Gabon | 0.50 | −1.59 | 2.48 | 1.04 |
| Congo | 0.44 | −1.61 | 2.41 | 1.03 |
| Nigeria | 0.41 | −1.71 | 2.34 | 1.03 |
| Sudan | 0.27 | −1.34 | 1.79 | 0.79 |
| Tunisia[†] | −0.16 | −3.42 | 3.03 | 1.64 |
| Benin[†] | −0.17 | −2.04 | 1.68 | 0.95 |
| Burkina Faso[†] | −0.19 | −2.09 | 1.63 | 0.95 |
| Morocco[†] | −0.36 | −3.40 | 2.64 | 1.55 |
| Algeria[†] | −0.44 | −2.49 | 1.50 | 1.02 |
| The Gambia[†] | −0.52 | −3.48 | 2.29 | 1.48 |
| Lesotho[†] | −0.55 | −3.48 | 2.23 | 1.47 |
| Equatorial Guinea[†] | −0.56 | −3.37 | 2.28 | 1.46 |

| Table 1 (continued) | | | | |
|---|---|---|---|---|
| Name | Mean | HDI low | HDI high | SD |
| Djibouti[†] | −0.58 | −3.50 | 2.10 | 1.44 |
| Egypt[†] | −0.70 | −3.57 | 1.92 | 1.42 |
| Mauritania[†] | −0.70 | −3.51 | 1.96 | 1.42 |
| Libya[†] | −0.74 | −3.61 | 2.00 | 1.44 |
| Rwanda[†] | −0.76 | −3.56 | 1.86 | 1.40 |
| Burundi[†] | −0.76 | −3.54 | 1.85 | 1.40 |
| Sierra Leone[†] | −0.77 | −3.62 | 1.77 | 1.39 |
| Guinea-Bissau[†] | −0.77 | −3.56 | 1.82 | 1.39 |
| Liberia[†] | −0.77 | −3.59 | 1.81 | 1.39 |
| Togo[†] | −0.80 | −3.53 | 1.83 | 1.39 |
| Eritrea[†] | −0.82 | −3.54 | 1.78 | 1.38 |
| Ghana[†] | −0.82 | −3.57 | 1.72 | 1.37 |
| Guinea[†] | −0.84 | −3.62 | 1.65 | 1.37 |
| Cote d'Ivoire[†] | −0.85 | −3.58 | 1.68 | 1.37 |
| Eswatini[†] | −0.86 | −3.65 | 1.67 | 1.37 |
| Somalia[†] | −1.04 | −3.67 | 1.38 | 1.31 |
| Nigeria[†] | −1.13 | −3.69 | 1.26 | 1.29 |
| Mali[†] | −1.14 | −3.74 | 1.26 | 1.30 |
| Niger[†] | −1.14 | −3.79 | 1.20 | 1.29 |
| Chad[†] | −1.42 | −3.91 | 0.85 | 1.23 |
| South Sudan[†] | −1.42 | −3.91 | 0.84 | 1.24 |
| DRC[†] | −1.57 | −4.05 | 0.59 | 1.21 |
| Angola[†] | −1.72 | −4.09 | 0.47 | 1.18 |

**Notes:**
HDI, Highest Density Interval; SD, standard deviation; Range, country range, based on IUCN Red List polygons (*IUCN, 2020*).
[*] Value suggests significant positive effect.
[†] Value suggests significant negative effect.

**Table 2 Regional geographical trends in large carnivore research in Africa (2000–2020), all species combined.** All figures were obtained by combining estimates of geographical range, studies/estimates, and density of studies/estimates for all large carnivore species. See Table S1 for species-specific insights.

| Region | Geographical range (km²)[*] | Studies | Density estimates | Studies/100,000 km² | Estimates/100,000 km² |
|---|---|---|---|---|---|
| Southern | 12,114,919 | 62 | 176 | 0.51 | 1.45 |
| Eastern | 12,587,456 | 38 | 86 | 0.30 | 0.68 |
| Central | 6,116,580 | 7 | 40 | 0.11 | 0.65 |
| Western | 4,832,540 | 5 | 8 | 0.10 | 0.17 |
| Northern | 8,145,160 | 1 | 1 | 0.01 | 0.01 |

**Note:**
[*] Based on IUCN Red List geographical range polygons (*IUCN, 2020*).

The greatest number of published lion population assessments came from eastern Africa, followed by southern, central, and finally western Africa (see Table S1 for regional-level species-specific insights). Nevertheless, western Africa exhibited the highest

density of lion assessments, due to the species' limited range in the region. Most leopard population assessments were carried out in southern Africa, with more than twice the number of studies than in all other regions combined. Cheetah population assessments also primarily took place in southern Africa, followed by eastern (which exhibited the greatest study density) and northern Africa, with none in western or central Africa. African wild dog assessments were relatively evenly spread across the continent. Most spotted hyaena population estimates were from southern Africa (also exhibiting the greatest study density), followed by eastern and central Africa; no studies took place in western or northern Africa. All three of the published striped hyaena population assessments took place in eastern Africa. Finally, brown hyaena assessments were only carried out in Southern Africa, the only region where the species is present.

### Country

South Africa was the country with the greatest total number of large carnivore population assessments, followed by Tanzania, Kenya, and Botswana. Study density, instead, was highest in Niger, followed by South Africa, Kenya, and Cameroon. Twenty-seven countries with at least one large carnivore species (57%) had no published density estimation studies (Table 3). Results from the GLMM Model 1 confirmed these results (Table S9). When controlling for range (GLMM Model 2), South Africa then Kenya still exhibited the strongest bias, while Tanzania dropped to fifth, with Cameroon and Botswana ranking higher (Table 1; Fig. 5). Chad, South Sudan, Democratic Republic of Congo (DRC), and Angola ranked lowest, due to the combination of large country range and a lack of studies (Table 3; Fig. 5).

Across all species, geographical range had a significant positive effect (GLMM Model 2; Table 1), indicating that having more species range in a country was associated with more studies. For individual species (GLMs), geographical range had a significant positive effect on the number of lion, leopard, and spotted hyaena population assessments within a country, but not on those of cheetah (Table 4; Tables S10–S12).

For individual species, Tanzania was the country with the greatest number of lion assessments, followed by Kenya and South Africa. Uganda was the country with the highest lion study density; if we only consider countries with >40,000 km$^2$ of lion range, however, South Africa exhibited the greatest study density, followed by Kenya, Zimbabwe, and Zambia. Central African Republic (hereafter CAR), Ethiopia, Namibia, and Mozambique all exhibited large species range, but few studies. Six countries (24%) had no published lion population assessments (Table S2). The GLM results revealed that Kenya, South Africa, and Cameroon had significantly more lion population assessments than would be expected based on country range, while CAR and South Sudan the fewest (Table 4).

For leopard, South Africa was the country with the most assessments by a wide margin (almost half of all studies). Only 28% of leopard range states had a published estimate, and the two with the greatest leopard range in Africa, Angola and the DRC, had none (Table S3). When controlling for geographical range in the GLM, South Africa was an outlier, exhibiting the largest positive bias in studies, and Zimbabwe, Cameroon and Kenya

**Table 3 Large carnivore population assessments in Africa (2000–2020) by country, all species combined.** Figures were obtained by combining estimates of geographical range, studies/estimates, and density of studies/estimates for all large carnivore species. Refer to Tables S2–S8 for species-specific insights.

| Country | Studies | Estimates | Geographical range (km$^2$)* | Studies/ 100,000 km$^2$ | Estimates/ 100,000 km$^2$ |
|---|---|---|---|---|---|
| South Africa | 27 | 88 | 1,601,900 | 1.69 | 5.49 |
| Tanzania | 16 | 41 | 2,871,700 | 0.56 | 1.43 |
| Kenya | 15 | 31 | 1,735,300 | 0.86 | 1.79 |
| Botswana | 11 | 38 | 2,469,150 | 0.45 | 1.54 |
| Namibia | 10 | 16 | 2,388,600 | 0.42 | 0.67 |
| Zambia | 7 | 12 | 1,305,900 | 0.54 | 0.92 |
| Cameroon | 5 | 29 | 600,740 | 0.83 | 4.83 |
| Zimbabwe | 4 | 13 | 752,800 | 0.53 | 1.73 |
| Ethiopia | 4 | 5 | 2,884,486 | 0.14 | 0.17 |
| Mozambique | 3 | 4 | 1,413,300 | 0.21 | 0.28 |
| Uganda | 2 | 7 | 468,900 | 0.43 | 1.49 |
| Senegal | 2 | 2 | 349,800 | 0.57 | 0.57 |
| CAR | 1 | 10 | 1,147,040 | 0.09 | 0.87 |
| Malawi | 1 | 6 | 135,600 | 0.73 | 4.42 |
| Gabon | 1 | 3 | 255,600 | 0.39 | 1.17 |
| Nigeria | 1 | 2 | 1,012,000 | 0.10 | 0.20 |
| Sudan | 1 | 2 | 1,729,400 | 0.06 | 0.12 |
| Niger | 1 | 1 | 17,190 | 5.82 | 5.82 |
| Benin | 1 | 1 | 300,400 | 0.33 | 0.33 |
| Congo | 1 | 1 | 386,000 | 0.26 | 0.26 |
| Burkina Faso | 1 | 1 | 572,550 | 0.17 | 0.17 |
| Algeria | 1 | 1 | 3,089,900 | 0.03 | 0.03 |
| Angola | 0 | 0 | 2,009,200 | 0.00 | 0.00 |
| Burundi | 0 | 0 | 28,700 | 0.00 | 0.00 |
| Chad | 0 | 0 | 1,940,300 | 0.00 | 0.00 |
| Cote d'Ivoire | 0 | 0 | 273,400 | 0.00 | 0.00 |
| Djibouti | 0 | 0 | 24,800 | 0.00 | 0.00 |
| DRC | 0 | 0 | 1,751,000 | 0.00 | 0.00 |
| Egypt | 0 | 0 | 1,015,800 | 0.00 | 0.00 |
| Equatorial Guinea | 0 | 0 | 12,800 | 0.00 | 0.00 |
| Eritrea | 0 | 0 | 257,798 | 0.00 | 0.00 |
| Eswatini | 0 | 0 | 38,369 | 0.00 | 0.00 |
| Ghana | 0 | 0 | 209,100 | 0.00 | 0.00 |
| Guinea | 0 | 0 | 277,100 | 0.00 | 0.00 |
| Guinea-Bissau | 0 | 0 | 43,000 | 0.00 | 0.00 |
| Lesotho | 0 | 0 | 100 | 0.00 | 0.00 |
| Liberia | 0 | 0 | 41,200 | 0.00 | 0.00 |
| Libya | 0 | 0 | 1,758,000 | 0.00 | 0.00 |

(Continued)

| Table 3 (continued) | | | | | |
| --- | --- | --- | --- | --- | --- |
| Country | Studies | Estimates | Geographical range (km²)* | Studies/ 100,000 km² | Estimates/ 100,000 km² |
| Mali | 0 | 0 | 1,602,600 | 0.00 | 0.00 |
| Mauritania | 0 | 0 | 1,140,960 | 0.00 | 0.00 |
| Morocco | 0 | 0 | 976,900 | 0.00 | 0.00 |
| Rwanda | 0 | 0 | 28,500 | 0.00 | 0.00 |
| Sierra Leone | 0 | 0 | 62,700 | 0.00 | 0.00 |
| Somalia | 0 | 0 | 1,314,700 | 0.00 | 0.00 |
| South Sudan | 0 | 0 | 1,271,872 | 0.00 | 0.00 |
| The Gambia | 0 | 0 | 11,300 | 0.00 | 0.00 |
| Togo | 0 | 0 | 54,600 | 0.00 | 0.00 |
| Tunisia | 0 | 0 | 163,600 | 0.00 | 0.00 |

**Note:**
* Based on IUCN Red List geographical range polygons (*IUCN, 2020*).

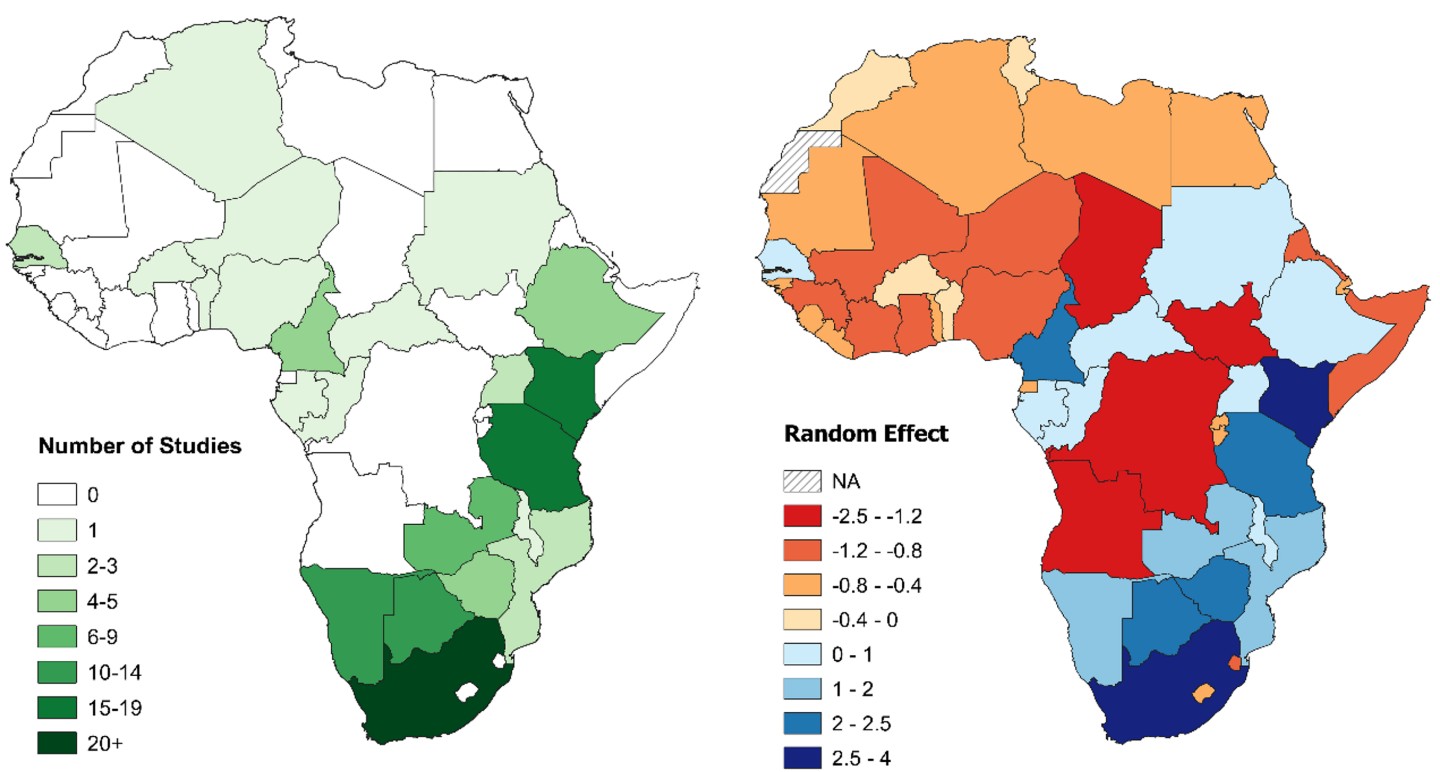

**Figure 5 Number of large carnivore population assessment studies in Africa and random effects for individual countries.** Left: number of large carnivore population assessment studies in Africa, by country. Right: random effects for individual countries from the Poisson GLMM investigating biases in large carnivore population assessments in Africa, accounting for differences in geographical range between countries (Model 2). A positive value (blue) indicates more population assessments than expected based on large carnivore geographical range within the country, while a negative value (red) fewer (all species combined). See Table 1 for country specific values. For all figures in this study, country boundaries are based on the definitions of the African Union (https://web.archive.org/web/20130927110741/http://www.afrimap.org/english/images/report/AfriMAP-AU-Guide-EN.pdf). 

**Table 4 Results of the models investigating biases in lion population assessments in Africa, accounting for geographical range (Negative Binomial GLM).** For residuals of individual countries, a positive value indicates more assessments than expected after controlling for the relevant variables, a negative value fewer. Significant residual values (defined as zero not contained within the 95% highest density interval (HDI) of the posterior distribution) are highlighted. Names of fixed effects and of countries are in bold if the variable exhibits a significant (*i.e.*, credible non-zero) effect. Similar results for other species are presented in Appendix S4.

| Name | Mean | HDI low | HDI high |
|---|---|---|---|
| (Intercept) | 0.73 | 0.29 | 1.17 |
| **Range**\* | **0.51** | **0.08** | **0.95** |
| **Kenya**\* | **3.46** | **1.59** | **5.36** |
| **South Africa**\* | **2.35** | **0.94** | **3.73** |
| **Cameroon**\* | **2.05** | **0.78** | **3.36** |
| Zimbabwe | 0.66 | −0.01 | 1.34 |
| Zambia | 0.62 | −0.08 | 1.33 |
| Uganda | 0.36 | −0.25 | 0.98 |
| Botswana | 0.35 | −0.50 | 1.21 |
| Tanzania | 0.28 | −1.09 | 2.12 |
| Nigeria | −0.29 | −0.67 | 0.10 |
| Sudan | −0.3 | −0.67 | 0.10 |
| Burkina Faso | −0.31 | −0.68 | 0.08 |
| Benin | −0.31 | −0.68 | 0.07 |
| Senegal | −0.33 | −0.69 | 0.04 |
| **Mozambique**† | **−0.52** | **−0.96** | **−0.08** |
| **Ethiopia**† | **−0.63** | **−0.95** | **−0.31** |
| **Namibia**† | **−0.68** | **−1.02** | **−0.35** |
| **Niger**† | **−0.91** | **−1.21** | **−0.63** |
| **Chad**† | **−0.91** | **−1.21** | **−0.62** |
| **Malawi**† | **−0.92** | **−1.22** | **−0.63** |
| **Angola**† | **−0.92** | **−1.23** | **−0.63** |
| **DRC**† | **−0.92** | **−1.23** | **−0.64** |
| **South Sudan**† | **−0.93** | **−1.24** | **−0.64** |
| **CAR**† | **−0.99** | **−1.57** | **−0.49** |

Note:
HDI, Highest Density Interval; Range, Geographical range of species, based on IUCN Red List range maps (*IUCN, 2020*).
\* value suggests significant positive effect;
† value suggests significant negative effect.

also had significantly more assessments than would be expected based on geographical range. Angola, DRC, and Ethiopia were the most understudied (Table S10).

Botswana and Kenya were the countries with the most cheetah assessments, followed by Namibia and Tanzania. 60% of cheetah range states had no published assessments, including some with large tracts of range (Chad, Ethiopia; Table S4). Kenya, Botswana, and Tanzania were the countries with the strongest positive bias after controlling for range in the GLM, while Algeria, Chad, and Ethiopia those exhibiting the strongest negative

research bias (Table S11). Peer-reviewed African wild dog population assessments were only carried out in four countries: South Africa, Kenya, Zimbabwe, and CAR. Study density was highest in South Africa. Overall, 78% of range countries had no published African wild dog population assessments, including the five countries with the largest geographical range (Table S5).

South Africa, Cameroon, Ethiopia, Kenya, and Namibia had the most spotted hyaena studies. Cameroon had the highest study density, followed by South Africa and Congo. A total of 24 range countries (63%) had no studies (Table S6), including Angola and DRC, the countries with the first and third greatest species range. Cameroon, South Africa, and Namibia exhibited the largest values of the random intercepts when accounting for geographical range in the GLM, and Angola, DRC, and Nigeria the smallest (Table S12). For striped hyaena, peer-reviewed assessments have only taken place in Kenya, with 96% of range states having no published density estimates (Table S7). Finally, South Africa then Botswana were the countries with the most brown hyaena population assessments, and Botswana, Zimbabwe, and South Africa those with the greatest study density. Four range countries (50%) had no studies (Table S8).

# DISCUSSION

## Taxonomic representativeness and biases

Over the past two decades, lion is the species that has received the greatest number of population assessments, both overall (Fig. 2) and relative to species' geographical range (Table 1). It is likely that the lion's highly charismatic nature (Macdonald et al., 2015), its role as a keystone and flagship species (Bauer et al., 2015b; Leader-Williams & Dublin, 2000), and its ability to attract conservation funding and generate revenue through tourism (Maciejewski & Kerley, 2014) all play a role. In addition, a variety of methodologies have been employed to survey lion populations, including some which cannot typically be applied to most other large carnivore species due to behavioural differences (e.g., 'call-ins'; Braczkowski et al., 2020). This likely also played a part in the species receiving greater research attention.

The large number of studies on spotted hyaena, the second most studied species, are instead likely partly driven by its vast distribution (as indicated by our findings; Fig. 4), and by the relatively large number of methods used to survey the species (Davis et al., 2022). In addition, assessments primarily targeting lions have often employed the data collected to also estimate spotted hyaena density (e.g., Ferreira & Funston, 2016). Nevertheless, the fact that—due to the species' wide distribution—study density was still relatively low (Fig. 3) suggests that considerable knowledge gaps remain across its range, especially in western and central Africa.

Striped hyaena had the lowest number of population assessments, and the strongest negative research bias (Table 1; Fig. 4). This is in line with previous suggestions that knowledge of the species is particularly low (AbiSaid & Dloniak, 2015), and is likely due to a combination of factors, including: populations often existing outside of formally protected areas (PAs; AbiSaid & Dloniak, 2015); the species being more secretive and less well-known (Macdonald et al., 2015), and thus less apt at raising conservation and tourism

revenue (*Di Minin et al., 2013*; *Okello, Manka & D'Amour, 2008*); and the fact that much of its range is in northern Africa, a region exhibiting low levels of conservation research in general (*Agha et al., 2018*; *Hickisch et al., 2019*; *Trimble & van Aarde, 2012*). Finally, the fact that some easily implementable methods used to survey other species (*e.g.*, call-ins) cannot be reliably applied to striped hyaena (*AbiSaid & Dloniak, 2015*) is likely to also have played a part.

The other species that exhibited a negative bias in research was the African wild dog (although this effect was not significant; Table 1, Fig. 4). This is of particular concern due to the species being the most threatened of the African large carnivores (*Woodroffe & Sillero-Zubiri, 2020*), and supports suggestions that conservation research can be poorly aligned with conservation priorities (*Di Marco et al., 2017*; *Wilson et al., 2016*). The observed paucity of estimates for African wild dog is likely a result of the species being less well-known compared to the felids (*Macdonald et al., 2015*), and a combination of ecological and methodological factors. African wild dogs are social, low density species, with very large home ranges that often range outside of PA boundaries (*Creel & Creel, 2002*); as a result, survey methods often employed for other species (*e.g.*, camera trapping combined with capture-recapture modelling; *Strampelli et al., 2022b*) are challenging to apply to African wild dogs, while others (*e.g.*, call-ins) are less suitable due to behavioural characteristics. Indeed, although rapid assessment methods such as spoor counts have occasionally been employed to survey populations (*Henschel et al., 2020*), approximately half of published density estimates were obtained through resource-intensive long-term studies or citizen-science approaches, highlighting the difficulty associated with surveying the species rapidly. In addition, most African wild dog populations in South Africa are part of an intensively managed metapopulation (*Nicholson et al., 2020*) where the exact number of individuals is known, and were therefore not eligible for our review, even though these populations are actively monitored.

Other species experienced weaker effects. Leopard exhibited a slight positive bias in research, while there was little evidence of cheetah experiencing strong biases in either direction. The suitability of leopard to camera trap (*Searle et al., 2021*) and sign-based (*Henschel et al., 2020*) methods has facilitated assessments, as has its wide range and charismatic nature (*Maciejewski & Kerley, 2014*). On the other hand, although cheetahs are highly charismatic (*Di Minin et al., 2013*; *Macdonald et al., 2015*), listed as vulnerable to extinction (*Durant et al., 2015*), and have high potential to generate conservation revenue through tourism (*Maciejewski & Kerley, 2014*; *Okello, Manka & D'Amour, 2008*), a lack of rapid survey techniques for the species (*Strampelli et al., 2021*) likely played a role in survey efforts not being greater. Additionally, the fact that—as for African wild dog—numerous populations in South Africa inhabit small, intensively managed reserves (*Buk et al., 2018*) also likely played a part in the estimates considered being relatively few.

For all species except cheetah, greater geographical range in a country was found to correspond to greater research effort in that country (Table 4; Tables S10–S12). Nevertheless, our findings suggest that a range of additional factors, including the species' ecology, charisma, ability to generate conservation funding and tourism revenues, and

applicability of different survey methods all play a role in driving the extent of population assessments for a species.

## Regional trends and biases

Greater research effort in southern Africa (Table 2) is likely a result of greater conservation investments in the region (*Brockington & Scholfield, 2010*; *Wilson et al., 2016*) and the better socio-economic status of some countries (*UNDP, 2020*). For most species, eastern Africa was the second most studied region; this is likely due to the region still harbouring numerous important large carnivore populations (Table S1), as well as the long history of conservation investments and research in some countries (*e.g.*, Kenya, Tanzania; *Brockington & Scholfield, 2010*).

Low research effort in northern Africa is likely due to most large carnivore populations in the region existing at low density and outside of PAs (*Belbachir et al., 2015*), to conservation research investments being comparatively low, and to the political instability of some countries (*Di Marco et al., 2017*; *Wilson et al., 2016*). Similarly, the low level of research in northern, western, and central Africa for most species (Table S1) mirrors the region's under-representation across wider conservation research (*Hickisch et al., 2019*; *Trimble & van Aarde, 2012*; *Wilson et al., 2016*). For central Africa, this negative bias is likely primarily a consequence of high volatility and low conservation investments in the region (*Brockington & Scholfield, 2010*; *FFP, 2020*). In western Africa, the fact that conservation investments and wildlife-oriented tourism have historically been lower than in southern and eastern Africa (*Brockington & Scholfield, 2010*; *UNWTO, 2018*) has likely had an impact on research effort.

Our findings largely agree with suggestions of biases towards carnivore research and conservation activities in eastern and southern Africa (*Ray, Hunter & Zigouris, 2005*). Nevertheless, when accounting for greater geographical range in these region, the density of population assessments was actually lower than in central and western Africa for a number of species (Table S1). In addition, for several species, many assessments in eastern Africa are from a small number of populations; for example, approximately half (48%) of lion and 75% of cheetah assessments in the region were from a single population, in the Serengeti-Mara (Appendix S2), even though this comprises only ~7% of lion's and ~8% of cheetah's eastern African range. Without studies from the Serengeti-Mara, eastern Africa would exhibit the lowest lion study density of any range region. Our findings therefore suggest that after accounting for differences in geographical range and the repeated sampling of a few populations, the observed biases in population assessments towards certain regions are less clear-cut than they may initially appear, and that understudied populations remain even in the regions receiving the most research attention.

## Country trends and biases

As lack of research can impede biodiversity conservation (*Di Marco et al., 2017*; *Pullin et al., 2004*), the fact that 57% of African countries with large carnivore range did not have a single published, peer-reviewed population assessment is of concern. When accounting for differences in geographical range, Angola, South Sudan, DRC, and Chad exhibited the

strongest negative bias in research (Table 1), likely primarily as a consequence of years of political unrest and insecurity in these countries (*FFP, 2020*).

South Africa showed the strongest positive bias in published research (Table 1), with the greatest number of assessments for several species (Tables S2–S8). Conservation investments are particularly high in South Africa (*Brockington & Scholfield, 2010*), likely explaining this bias. Although Tanzania was the country with the second greatest number of assessments, accounting for geographical range caused the country to exhibit only the fifth strongest positive bias (Table 1), indicating that part of this effect is due to the country being home to considerable large carnivore range. Furthermore, 60% of all published lion population assessments in the country were from a single ecosystem (Serengeti), comprising only 8% of lion country range. A similar bias was also observed in Kenya, with 45% of lion studies being from the Mara complex, even though this accounted for only ~3% of national lion range. These biases towards a small number of well-studied populations suggest that the total number of studies may not be a representative indicator of how well a country's large carnivore populations have been studied.

Overall, our results suggest that the extent of published biological research in a country is dependent on a range of factors, including socio-economic status, research history and interests of individuals and organisations, priorities of funding agencies and governments, in-country support and capacity, and language barriers to publication (*Griffiths & Dos Santos, 2012*; *Trimble & van Aarde, 2012*).

## Opportunities and recommendations

Reducing the identified geographical and taxonomic biases in population assessments would help ensure that all species and areas of conservation importance have an adequate knowledge base available, improving their conservation outlook (*Wilson et al., 2016*). We argue that addressing the identified geographical biases (by region, countries, and land use types) should be the most pressing priority: while a focus towards certain species, such as lion, may indirectly lead to conservation benefits for other species (*e.g.*, by identifying areas requiring additional conservation investments), research from well-studied areas is unlikely to be able to help inform conservation decisions in poorly-studied regions (*Di Marco et al., 2017*). Thus, much like for the wider conservation research field (*Trimble & van Aarde, 2012*), geographical biases in research and assessments are more immediate hurdles for science-based conservation management of African large carnivores. As a result, northern, western, and central Africa should be considered priority regions for future research. Increased attention should in particular be given to the twenty-six countries which currently lack any published estimates (Table 3), especially Angola, DRC, South Sudan, and Chad, given their considerable large carnivore country ranges and their potential importance for the conservation of these species (*Dickman et al., 2015*). Within southern and eastern Africa, we encourage prioritising additional research outside of South Africa and the Serengeti-Mara ecosystem, respectively, to complement the efforts there.

Shortfalls in conservation research in countries with high levels of biodiversity but low income can be due to a lack of trained and/or funded biologists (*Gaston, 2000*), research infrastructure (*Wilson et al., 2016*), or dedicated finances (*Githiru et al., 2015*). As a result,

building capacity of researchers and practitioners in large carnivore survey and monitoring techniques in under-represented areas should be a priority. The fact that only 59% of studies outside of South Africa included a co-author from the study country reinforces suggestions that research in developing countries is disproportionately led by scientists from more developed areas (*Stocks et al., 2008*), and shows there is considerable need for such capacity building efforts. For these reasons, we recommend that both donors and foreign researchers maximise the involvement of local scientists, students, and practitioners in future assessments, including through capacity building initiatives such as the provision of training, funding, and equipment.

Conservation donors and funders should encourage efforts in understudied regions, as well as for understudied species, to ensure that conservation research occurs where it is most needed (*Di Marco et al., 2017*; *Wilson et al., 2016*). This is especially the case given widespread funding shortfalls in conservation (*Lindsey et al., 2018*). At the same time, we also urge funders and practitioners to recognise the importance of scale dependency, as even within more studied countries gaps remain: in Tanzania, the country with the greatest lion range and the most lion studies, most populations are nevertheless yet to be empirically assessed. We therefore emphasize the importance of decision and investment processes being multi-scale, and of appreciating the intricacies of the identified biases.

On a species level, we echo calls for further population assessments of striped hyaena (*AbiSaid & Dloniak, 2015*). We also strongly recommend prioritising further population assessments of African wild dog, particularly due to the species' classification as 'Endangered' (*Woodroffe & Sillero-Zubiri, 2020*). Such efforts are especially required in countries that have been identified as critical for the species, but where no recent assessments have been carried out (*e.g.*, Botswana and Tanzania; *Kuiper et al., 2018*). As the lack of well-established methods to rapidly survey African wild dog populations is a key reason for these knowledge gaps (*Woodroffe & Sillero-Zubiri, 2020*), longer-term intensive monitoring studies such as those carried out prior to this review's considered period (*e.g.*, *Creel & Creel, 1996*), alongside further development of novel, cost-effective methodologies (*e.g.*, citizen-science techniques; *Marnewick et al., 2014*), are strongly recommended. The possibility of monitoring populations through alternative status parameters, such as species' occupancy (*Henschel et al., 2020*), should also be explored.

Our findings also highlight the urgent need for additional cheetah population assessments, particularly in northern, western, and central Africa. Due to their large country ranges, studies in Chad and Ethiopia should especially be considered a priority. As in the case of African wild dog, we also recommend further development and standardisation of cheetah population monitoring techniques, including the exploration of citizen-science based approaches (*Marnewick et al., 2014*; *Weise et al., 2017*).

For leopard, surveys are particularly recommended in the 72% of range states without published estimates, particularly those with large geographical range (Angola, DRC, Ethiopia, and South Sudan; Table S3). Although considered a highly adaptable species, leopard populations are increasingly under threat (*Stein et al., 2020*). It is therefore important that research and monitoring is carried out across the species' range, rather than in localised 'hubs', as is currently the case (South Africa; Table S3).

For spotted hyaena, future efforts should prioritise populations in northern and western Africa, as no assessments are available from these regions, as well as in countries with extensive range but no surveys (Angola, DRC, Nigeria, Somalia, and South Sudan). For brown hyaena, we recommend the prioritisation of assessments across different habitats and land-use types (*Kent & Hill, 2013*), and in countries where the species is yet to be surveyed (Angola, Mozambique, Eswatini; Table S8).

Study density for lion was lowest in central Africa, which should therefore be considered a priority region for further assessments. Nevertheless, considerable opportunity for further work exists in most range countries. Overall, the fact that even for lion, the species with the strongest positive bias in research, considerable gaps still exist highlights the extent to which African large carnivore populations are presently understudied and under-monitored.

Finally, efforts should be made to address the observed sampling bias towards photographic tourism areas, and against trophy hunting areas. Trophy hunting areas in Africa cover a greater area than National Parks (*Lindsey, Roulet & Romañach, 2007*), and, given that accurate estimates of population size are crucial to ensure the sustainability of trophy hunting (*Mweetwa et al., 2018*), we recommend that future efforts attempt to bridge this gap. Similarly, although working on public and/or unprotected land can be logistically challenging (*Agha et al., 2018*), efforts should also be made to address the negative sampling bias associated with these areas, as our findings support the global pattern of biodiversity monitoring largely taking place within PA networks (*Martin, Blossey & Ellis, 2012*). The fact that some species still heavily occupy areas outside PAs (*e.g.*, spotted hyaena, *Bohm & Höner, 2015*; cheetah, *Weise et al., 2017*), and that non-protected areas encompass the majority of wildlife habitat across the continent (*Agha et al., 2018*), highlights the need for future efforts to include boundary and non-protected areas.

## CONCLUSIONS

We carried out the first review of peer-reviewed African large carnivore population assessments, focusing on the last two decades, and empirically tested for geographical and taxonomic biases in effort. We found research biases towards lion and against striped hyaena, and to a lesser extent African wild dog. Assessments were biased towards southern and eastern Africa, while northern, western, and central Africa were generally under-studied. Non-protected and trophy hunting areas were under-sampled compared to photographic tourism areas. Significant opportunities exist for greater inclusion of host country nationals in such studies, and we recommend increased capacity building and provision of funding for researchers from range countries. Overall, we recommend that the biases we have identified are employed by researchers, practitioners, and policymakers to address knowledge gaps and help inform future research and monitoring efforts.

## ACKNOWLEDGEMENTS

We thank Prof. E. J. Milner-Gulland and Dr. Luke L. Hunter for the insightful comments on the study.

### Funding

Scholarship funding for Paolo Strampelli was provided by the University of Oxford's NERC Environmental Research DTP, and by The Queen's College of the University of Oxford. Amy J. Dickman was funded by a Recanati-Kaplan Fellowship. The funders had no role in study design, data collection and analysis, decision to publish, or preparation of the manuscript.

### Grant Disclosures

The following grant information was disclosed by the authors:
University of Oxford's NERC Environmental Research DTP.
The Queen's College of the University of Oxford.
Recanati-Kaplan Fellowship.

### Competing Interests

The authors declare that they have no competing interests.

### Author Contributions

- Paolo Strampelli conceived and designed the experiments, performed the experiments, analyzed the data, prepared figures and/or tables, authored or reviewed drafts of the article, and approved the final draft.
- Liz A. D. Campbell conceived and designed the experiments, performed the experiments, analyzed the data, authored or reviewed drafts of the article, and approved the final draft.
- Philipp Henschel conceived and designed the experiments, authored or reviewed drafts of the article, and approved the final draft.
- Samantha K. Nicholson performed the experiments, authored or reviewed drafts of the article, and approved the final draft.
- David W. Macdonald conceived and designed the experiments, authored or reviewed drafts of the article, and approved the final draft.
- Amy J. Dickman conceived and designed the experiments, authored or reviewed drafts of the article, and approved the final draft.

### Data Availability

The data is available at GitHub: https://github.com/pstrampelli/African-Large-Carnivore-Density-Studies-Estimates.

### Supplemental Information

Supplemental information for this article can be found online at http://dx.doi.org/10.7717/peerj.14354#supplemental-information.

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
