# Peer review of "Trends and biases in African large carnivore population assessments: identifying priorities and opportunities from a systematic review of two decades of research"

_PeerJ, doi:10.7717/peerj.14354_

## Round 0.1 · original submission · Major Revisions

Thank you for submitting this interesting study to PeerJ. The manuscript requires substantial revisions before we can consider whether it might be suitable for publication. The reviewers have provided many useful comments that need to be addressed in a new revised version. One of the reviewers also submitted an annotated pdf file. Please provide comprehensive responses to all the comments.

The paragraph structure in the Introduction could be improved (see Lines 58, 76, 93). This paper contains lots of useful advice: "The art of writing science" https://doi.org/10.1002/pro.514 e.g. "Conceptually, a paragraph should also stand on its own two feet. For example, the first sentence of a paragraph generally should not refer to an idea in the paragraph above without, at the least, restating that idea."

Reviewer 1 ·

Excellent Review

This review has been rated excellent by staff (in the top 15% of reviews)
EDITOR COMMENT
Thanks very much for this comprehensive and insightful review.

Basic reporting

The manuscript is well-written with a strong command of English. It is well-structured and polished. The authors are recognized experts in the field of African large carnivore ecology and conservation and did an overall solid job. It also is an important topic that highlights several issues and knowledge gaps.

Most referencing is done well, though I think the reference list is a bit short.

My comments are extensive and commensurate with major revisions.

Experimental design

The study is comprised of original research and fits the aims and scope of the journal. The topic is well-defined and relevant and clearly states the knowledge gap. The investigation is quite rigorous and the methods well-described, for the most part. Issues are listed below; none are deal breakers but deserve consideration.

1. 20-year window: I wonder why you "only" chose a 20-year window, considering how much research was done from the 1960s-1990s on large carnivores in Africa. You mention several times that you wanted to investigate trends over "two decades" but do not clarify why, or if there is any reason it was not restricted to one decade or expanded to three or more. Some more justification would be appreciated. After all, you might have picked up more trends with an even larger time scale.

2. Inclusion of studies employing spurious methodologies: I appreciate you being forthright about the debates surrounding spoor counts, call-ups, SECR, etc. for large carnivore density estimates. However, you say your goal was to "assess research effort" and that you "did not make distinctions based on methods employed", yet you only considered peer-reviewed sources. This somehow seems inconsistent and contradictory; there are, after all, robust methodologies such as SECR employed in pre-prints, theses, and government reports. I do not fully understand why these were discarded and feel your argument about “[ensuring] data and information quality” to be weak even though you mention it in detail in your SI. After all, peer review is far from perfectly critical and does NOT ensure quality. There is a lot of bad science out there... can you justify your decision better and more consistently?

3. Ranges used: The authors should acknowledge the fact that the ranges they used are not always accurate. This is especially true for the three hyena species included in this study; the last full range/distributional assessment for the Hyaenidae was completed in 1998 using outdated methods. There is an ongoing effort to update the ranges; it is not yet done, but the new ranges will be slightly different than the ones you mapped out for your study. For example – that bizarre “island” without spotted hyenas in Zimbabwe is inaccurate. For leopards too, how reliable are range estimates in politically unstable countries in e.g., Central Africa? I suggest you mention this issue in your limitations.

4. Bayesian approach: I found it a bit curious that you used Bayesian statistics (runjags) and MCMC chains as opposed to a frequentist approach. I usually see and run GLMMs in frequentist packages such as spaMM, for which you can easily use bootstrapping to get accurate p-values. Is there a reason why you did not use a frequentist approach? By the way, you wrote MCMC without explicitly defining/expanding the acronym the first time.

Validity of the findings

The data are available for download on GitHub and are robust and sound. The results are reproducible. The conclusions are justified and credible, and well-linked to the main story of the manuscript.

Additional comments

Lines 46-49: the species seem to be listed somewhat randomly - I can see you listed felids first, then a canid, then hyenas, but within those "sub-lists" they should be listed alphabetically or using some other systematic approach. Just seems odd otherwise.

Lines 51-57: this is a huge, convoluted, run-on sentence. Please split into 2-3 sentences. I had to read it several times repeatedly to even understand what was being said.

Line 63: “range-wide priorities” sounds odd… priorities related to what? Conservation?

Line 65: hm, one could argue population SIZE is even more of a “gold standard” than population density. Population size is used for IUCN classifications and has major implications for inbreeding and population persistence. A population might be very dense but consist of just a few individuals if the area is small.

Line 76: not all large carnivores occur at low densities (e.g., Ngorongoro Crater spotted hyenas occur at a density of >1/km2) nor are all nocturnal and secretive (e.g., the Serengeti cheetahs).

Line 78: it remains unclear to me WHY it is important to study and conserve large carnivores. Yes, there is a “moral” or “intrinsic value” argument to be made, but I (and I suspect much of the general public) find it underwhelming. Why not mention the practical benefits and challenges African large carnivores provide in terms of ecosystem services (https://doi.org/10.7717/peerj.7916), tourism revenue (https://doi.org/10.1016/j.biocon.2009.06.024), cultural importance (https://doi.org/10.3389/fcosc.2021.691975), livestock depredation (https://doi.org/10.1111/j.1523-1739.2003.00061.x), man eating (https://doi.org/10.2982/0012-8317(2001)90[1:TSOMAL]2.0.CO;2) etc.? All these topics make it important to get accurate density/status estimates for large carnivores due to their sheer importance (good and bad) to human societies.

Line 89: “track re-align”? What does that mean? I think you have a missing word.

Lines 93-101: this is a great final paragraph, well done.

Lines 111-112: for the hyenas, did you include both “hyena” (U.S. spelling) and “hyaena”?

Line 143: period missing at end of sentence.

Line 172: difference, not different.

Line 183: standardized how, exactly? Centered at 0? Transformed?

Line 184: ok, but you need to say what the VIF cutoff was. 3? 5? 10? What correlation coefficient was deemed problematic? 0.5? You can’t just say “…which found no issues”. There need to be explicit criteria used.

Line 191: GLMs, not GLM.

Line 198: you should say WHY Poisson was your default – I assume because it was discrete count data. But say that. Not everyone knows much about the different families and how the data needs to be characterized to be used.

Line 203: why did you capitalize negative binomial here but not in the previous sentence?

Line 209: What is an “African large African species”?!

Line 240: I think the correct phrase here would be “inclusive of all species” rather than “across”. “across” makes the sentence sound contradictory with the next one.

Line 252: “all the only three” sounds awkward – I suggest rephrasing to “all three of the published…”

Line 262: no Oxford comma here but you tend to use it elsewhere in the manuscript – please be consistent.

Line 312: just say “a variety of methodologies”

Line 314: hm, but spotted hyenas can be surveyed the exact same ways.

Line 316-322: another reason for the large number of studies on spotted hyenas may be that there are two very long-term scientific research projects exclusively dedicated to spotted hyenas: one at Michigan State University (USA) focused on the Masai Mara, Kenya and another by Leibniz-IZW (Germany) focused on Ngorongoro and Serengeti, Tanzania. Yet, most research on spotted hyenas has been dominated by behavioral and evolutionary ecology as opposed to anything conservation or management related, hence maybe why you found low study density.

Line 322: especially in Central and West Africa.

Line 334: shouldn’t you call it African wild dog to be consistent?

Line 337: well, a lot of wild dog research is focused on other topics such as metapopulation management and intraguild interactions as a “fugitive” species. It doesn’t mean it’s misaligned. It just so happens that wild dogs are strongly affected by the presence of lions, so maybe it makes sense to focus on intraguild interactions for this species in particular.

Lines 352-362: it’s a bit odd to me again that you mention leopards and cheetahs last here, but previously mention them second and third in other parts of the manuscript… the ordering is just all over the place.

Lines 392-394: and mostly in the open grassland areas at that, rather than the Serengeti woodlands or highland forests of Ngorongoro.

Line 404: this is the first time "DRC" is mentioned in the text but the full name is not spelled out. And again, the sequence of countries listed here seems to lack any semblance of order.

Line 413: you already mentioned this.

Line 414: “Mara system” sounds odd.

Line 427: it does not make sense to say “both” and then list three rather than two items.

Line 445-451: absolutely, well said.

Line 514: why capitalize northern?

References: ensure DOIs are included when available as well as page numbers etc. Please also make sure the reference list matches in-text citations.

Figures: please make sure all your color schemes are colorblind-friendly. Figure 5 is not.

Figures 1, 3, 4: the visibility for these figures seems awfully blurry; please ensure you upload files with at least 300 DPI or whatever the journal specifications are.

Figure 2: why are brown and striped hyenas combined into one map but all the other species get their own? Why did you choose a "dotted pattern" for only the brown hyena? You say you did it but did not state why.

Figure 3: why did you choose this particular ordering of the x-axis? Again, the ordering of lists and axes throughout the manuscript seems haphazard and sloppy. Please address this.

Table 4: columns are not aligned in the PDF.

SI: I may have missed it, but do you anywhere actually list which countries fall into which of your regional categories? For example, is Malawi eastern or southern? What about Somalia, is it northern or eastern?

Figure S2: missing space between “GLMM” and “The latter…”

Figure S5: why is the order of the the PPC and model fit panels for spotted hyena flipped compared to the panels for the other species?

·

Basic reporting

This study provides very important insight and draws much-needed attention to research gaps and biases in the literature on large carnivores of Africa.

Language and grammar are generally up to standard, professional, clear, and of high quality throughout the manuscript. I highlighted a couple of minor, potentially missed, errors or areas which could be revised for clarity under “General comments”.

The introduction provides concise, well-structured background information and valuable context for the need for such a review to be performed. Literature referenced generally covers a broad range and is applicable to the study. I support the use of broad-scale references where possible in the context of this study and believe that the authors covered this well.

The general structure of the study conforms to that expected of a review and to PeerJ standards. The structure is clear and logical. The abstract summarises the study well and sufficiently addresses all sections. Findings, as well as the rationale for the study, are clearly communicated and summarised within the abstract. The introduction provides a brief overview of the field, and the knowledge gap is addressed early on and in-depth in terms of the area, but less defined in terms of the study species. I would prefer a bit more detail about the importance of these large carnivores and the relevance of specifically focussing research efforts on them within the scope of the bigger picture (i.e. being umbrella/flagship species, large ranges etc.) in the opening paragraphs. In other words, why would you do this review for them but not necessarily for smaller mesopredators or perhaps large herbivores? Have similar reviews been done for other groups of species in Africa? I think that briefly referring to and addressing these points will also increase the readability for a wider audience, i.e. individuals who may not have a firm grasp on large carnivore literature in Africa yet and refer to a study such as this one to gain a broad-scale understanding. Any comments related to the methodology are addressed in the “Experimental design” section.

All figures and tables are relevant and well labelled/described. A PRISMA flow diagram is included as Figure 1 and a completed PRISMA checklist as Appendix S1, as is required by PeerJ policy. Nothing major but would make sense to keep the order in which graphs are presented in Appendix 5 consistent.

Raw data are supplied in Appendix 2 and are accessible.

Experimental design

This review study clearly fits the criteria and scope of PeerJ under the Environmental Sciences banner. I believe that it is important for this study to be open access as I consider it to be important in guiding future research efforts and should thus be readily available. Hence, in my opinion, this study is an ideal fit for PeerJ.

The aims of the study are well defined, relevant, and meaningful. The knowledge gap and need for the study are clearly communicated but see further suggestions under “Basic reporting” regarding providing sufficient context in terms of both species and area.

Methods are robust, well explained, of a high standard, and can be replicated if so required. It might be worth it to include a sentence or two more in the introduction that acknowledges the use of other population assessment methods, such as occupancy for example, which were not considered for the purposes of this study. Do you believe it would have made any difference to the outcomes if such studies were included? What are the problems or potential risks, if any, when including such studies (relative abundance…)?

Validity of the findings

Results are well stated.

The conclusions made by the authors are clear and answer the original research question. Inferences are limited to the spatial extent of the study (Africa). Findings are valid, required, and satisfy the criteria for publication.

Additional comments

Line #62 – I would not refer to data as “They”, but rather as “It” or perhaps simply “These data/information/knowledge/decisions”. It is not 100% clear to the reader whether you refer to the data or the management and policy decisions here.
Line #143 – Full stop required.
Line #209 – “African large African species”, which I assume should be African large carnivore species?
Line #252 – “The only three…all took place in..”
Line #313 “…be typically be…”, only one be?
Line #370 – “IN the region”
Line #378 – Political instability?
Line #398 – “even IN the regions”
Line #514 – “northern”
Line #517 – “nationals”
First sentence in Appendix S3 refers to Appendix S1 – I suspect it must be Appendix S2 instead.

Well done to the authors on a great manuscript and excellent work tackling a very important knowledge gap. I am sure that this paper will contribute significantly to future research prospects across Africa.

·

Basic reporting

Line 172: I think the author meant to write “difference”.
Line 209: Remove the word African before large. It should read as “large African species”.
Line 313: Remove “be” between typically and applied.
Line 369 -371: The sentence needs to be re-written. The message makes sense but the delivery is unclear.
Line 382: The sentence should read “is likely primarily a consequence”
Line 395 – 398: The information in this sentence is very important but the delivery loses the importance and relevance of the statement.

Experimental design

Line 156 – 158: Tracking and documenting the researcher nationality and affiliation may be challenging and bias. Perhaps if the inclusion of the associated institute could also add some context.

Line 348 – 351: Cheetah and wild dogs are generally well studied because of the metapopulation management. It would be helpful if the authors could justify why these were not considered as part of the analysis.

Validity of the findings

The statistical and modeling approach are sound and valid. Although the findings are not novel it is important that this information be documented and used as support for further investigation of the under studied species in challenging work environments.

Additional comments

The research results are very interesting. It would help if the authors could give more information or interpretation about the researcher nationality and affiliations. In the results this section is brief an glazed over. There is also the issue of funding projects particularly in Africa. This could also provide insights about preference or bias.

It is very interesting that hunting areas were not forth coming with information as they tend to have the resources for studies. It would also help if there was more of a distinction between monitoring and research.

---

## Round 0.2 · accepted · Accept

The reviewers are content with all the revisions.

Reviewer 1 ·

Basic reporting

No comment

Experimental design

No comment

Validity of the findings

No comment

Additional comments

Good work, I can see the manuscript was thoroughly revised and am satisfied with how you addressed the comments.

·

Basic reporting

The authors addressed all my initial concerns sufficiently. The restructuring of the introduction drastically improved clarity and readability.

Experimental design

The justification for the methods used is more clearly communicated in the manuscript. No further comments.

Validity of the findings

No further comments.

·

Basic reporting

The changes made to the manuscript allow for easier reading. The authors have provided clarity where necessary. The structure and order of the results are clearly shown. Outcomes from research such as this can be very beneficial when directing and guiding research and monitoring efforts.

Experimental design

The experimental design is appropriate and robust for the nature of the manuscript. There are no additional comments.

Validity of the findings

The findings are logical and clearly related to the questions being asked.

Additional comments

Generally the manuscript reads well. Clarity has been provided and where no changes were made, the authors provided satisfactory responses in the rebuttal. Although it is not the scope of this manuscript however the issue or concern with regards to political instability is relevant as this has been a documented factor in countries such as DRC and South Sudan. Although recommendations can be made, we must be realistic about country priorities.